# Identification of the Novel Oncogenic Role of SAAL1 and Its Therapeutic Potential in Hepatocellular Carcinoma

**DOI:** 10.3390/cancers12071843

**Published:** 2020-07-08

**Authors:** Pei-Yi Chu, Shiao-Lin Tung, Kuo-Wang Tsai, Fang-Ping Shen, Shih-Hsuan Chan

**Affiliations:** 1Department of Pathology, Show Chwan Memorial Hospital, Changhua 500, Taiwan; chu.peiyi@msa.hinet.net; 2School of Medicine, College of Medicine, Fu Jen Catholic University, New Taipei City 242, Taiwan; 3Department of Health Food, Chung Chou University of Science and Technology, Changhua 500, Taiwan; 4National Institute of Cancer Research, National Health Research Institute, Tainan 704, Taiwan; 5Department of Hematology and Oncology, Ton-Yen General Hospital, Hsinchu 302, Taiwan; sonoratung@gmail.com; 6Department of Nursing, Hsin Sheng Junior College of Medical Care and Management, Taoyuan City 325, Taiwan; 7Department of Research, Taipei Tzu Chi Hospital, Buddhist Tzu Chi Medical Foundation, New Taipei City 235, Taiwan; kwtsai6733@gmail.com; 8Graduate Institute of Integrated Medicine, China Medical University, No.91, Hsueh-Shih Road, Taichung 40402, Taiwan; a0939631309@gmail.com

**Keywords:** HCC, SAAL1, HGF, Met

## Abstract

Hepatocellular carcinoma (HCC) is the third leading cause of cancer deaths worldwide, affecting over 700,000 people per year. The treatment effect in advanced HCC is still disappointing and prognosis of advanced HCC remains poor. Hence, to find more effective therapeutic targets to improve the treatment outcome of HCC is of urgent need. In this study, we reported the novel oncogenic function of SAAL1 (serum amyloid A-like 1) in HCC, which previously is considered as an inflammation-related gene. We found that SAAL1 was significantly upregulated in HCC tumor tissues when compared to the adjacent normal tissues and high expression of SAAL1 correlated with shorter overall survival in The Cancer Genome Atlas (TCGA) HCC database. Functionally, we showed that the depletion of SAAL1 significantly reduced cell proliferation, 3D colony formation, and migration/invasion abilities of HCC cancer cells. Furthermore, suppression of SAAL1 impaired the HGF/Met-driven Akt/mTOR phosphorylation cascade and increased the chemosensitivity of HCC cells to sorafenib and foretinib treatment. Our data indicated that SAAL1 plays an important role in HCC via mediating oncogenic HGF/Met-driven Akt/mTOR signaling and could serve as an independent prognostic marker, as well as a promising therapeutic target for HCC patients.

## 1. Background

Hepatocellular carcinoma (HCC) is the third leading cause of cancer deaths worldwide, affecting over 700,000 people per year [1,2]. HCC comprises nearly 80% of the primary liver malignancies and occurs predominantly in the background of a cirrhotic liver and chronic liver diseases [1,2,3]. The known risk factors of HCC include viral infection (hepatitis B and hepatitis C viruses, etc.), cirrhotic liver, excessive alcohol intake, aflatoxins, obesity, diabetes, and non-alcoholic fatty liver disease, etc. [4,5]. Treatment of HCC mainly depends on the tumor stage, reservation of liver function, as well as the overall performance status of patients [6]. Nevertheless, more than 50% of HCC patients present an advanced disease upon diagnosis [7]. Curative surgical resection or liver transplantation or local therapy, such as transarterial chemoembolization (TACE) failed to show significant efficacy in prolonging overall survival in advanced HCC, the treatment effect of conventional chemotherapeutic agents such as doxorubicin in advanced HCC is also disappointing [8]. In spite of new targeted therapies and immunotherapeutic agents such as sorafenib, regorafenib, cabozantinib, lenvatinib, and nivolumab were applied to treat advanced HCC, the prognosis of advanced HCC remains poor with a median survival of less than two years [9,10,11,12,13]. Based on the above-mentioned facts, HCC is still one of the hard-to-treat human malignancies, and more effective therapeutic approaches to improve the treatment outcome of HCC are in an urgent need.

Serum amyloid A-like 1 (SAAL1) is a serum amyloid A (SAA) related gene and located in the same chromosome (11p) with SAA family genes [14]. SAAL1 is a novel acute phase responsive protein associated with the dysregulated proliferation of activated synovial fibroblasts to accelerate the progression of synovitis [15]. Upregulation of SAAL1 has been found in the foot joints of mice with collagen-induced arthritis and knockdown of SAAL1 can inhibit the proliferation of human rheumatoid arthritis synovial fibroblast (RASFs) in vitro [15]. Another study using Oplegnathus fasciatus as a model system revealed that the level of SAAL1 was significantly elevated in the blood and liver upon the challenge of bacterial infection [14]. Like the other two major acute-phase proteins SAA and C-reactive protein (CRP), SAAL1 protein expression has been found to be induced in the liver upon inflammation and could serve as a biomarker of the acute phase response [14]. Although SAA family genes were reported to promote tumor angiogenesis and metastasis in several cancers, the role of SAAL1 is unknown in cancer and HCC tumorigenesis [16,17].

In the present study, we explored the possible roles of SAAL1 in HCC. We are the first to show that the expression of SAAL1 was upregulated in HCC tumor tissues and correlated with poor overall survival in HCC patients. We demonstrated that depletion of SAAL1 in HCC cells led to the inhibition of cell proliferation and anchorage-independent growth, as well as migration/invasion abilities via impairing HGF/Met-driven Akt/mTOR oncogenic signaling. Moreover, inhibition of SAAL1 significantly increased the chemosensitivity of HCC cells towards sorafenib and foretinib treatment. Our data first revealed that SAAL1 is a potential prognostic biomarker of HCC and could serve as a promising therapeutic target in HCC treatment.

## 2. Results

### 2.1. SAAL1 Is Upregulated in HCC Tumor Tissues and Correlates with Poor Overall Survival in HCC Patients

We analyzed the expression status of SAAL1 in HCC using online databases. First, the analysis of TCGA RNAseq and GENT databases [18] showed that the expression of SAAL1 is significantly higher in HCC tumor tissues than that of the adjacent normal tissues (Figure 1A). For Kaplan-Meier survival analysis, we defined a cut-off value for SAAL1 expression levels using receiver-operating characteristics (ROC) analysis to separate 346 patients into two groups, which represent SAAL1 high (*n* = 151) and SAAL1 low groups (*n* = 195), respectively. Kaplan-Meier survival analysis showed that patients with higher SAAL1 expressions were significantly associated with the shorter overall survival than those patients with lower SAAL1 expressions (*p* = 0.009) (Figure 1B and Table 1). In addition, we found that there was no significant association between SAAL1 expression and HCC TNM stage (Appendix A). Univariate Cox’s regression analysis showed that high levels of SAAL1 resulted in poor overall survival of HCC patients (crude hazard ratio [CHR], 1.63; 95% confidence interval (CI), 1.13–2.35; *p* = 0.009). Multivariate analysis indicated that the expression of SAAL1 was an independent predictor for the poor prognosis of HCC patients (adjusted hazard ratio [AHR], 1.57; 95% confidence interval (CI), 1.09–2.27; *p* = 0.016). Taken together, we are the first to report that SAAL1 expression was upregulated in HCC and could be served as an independent prognostic marker for poor overall survival in HCC patients. These results indicate that SAAL1 may play an oncogenic role in HCC.

### 2.2. Depletion of SAAL1 Significantly Impairs HCC Cell Proliferation and Anchorage-Independent Growth via Inducing G1 Phase Cell Cycle Arrest

To explore the potential role of SAAL1 in HCC tumorigenesis, the effect of depletion of SAAL1 on tumor growth was analyzed. First, SAAL1 expression was depleted in three human HCC cells Hep-3B, SK-Hep1, and PLC/PRF5 by siRNAs transfection. The results showed that SAAL1 was significantly depleted at the mRNA and protein level, respectively, in three HCC cancer cell lines, Hep3B, SK-Hep1, and PLC/PRF5 using qRT-PCR and Western blot analysis (Figure 2A and Appendix A). Cell proliferation of the SAAL1 siRNA-transfected cells was examined for six days. The results showed that the depletion of SAAL1 significantly impaired cell proliferation compared to the control siRNA in three HCC lines (Figure 2B). Next, we investigated whether SAAL1 depletion would affect HCC cell growth in a three-dimensional (3D) setting. To do so, we applied a 3D Matrigel culture, which best recapitulates tumor growth in vivo, in SK-Hep1, PLC/PRF5, and Hep-3B lines and found that SAAL1 depletion greatly inhibited anchorage-independent growth in three HCC lines (Figure 2C,D).

Given that knockdown of SAAL1 expression reduced HCC cell proliferation and 3D colony growth, we next examined if the expression of cell cycle-related proteins were altered in the SAAL-depleted HCC cells compared to the control cells. The Western blot result showed that inhibition of SAAL1 expression led to the increase in p21 and p27 protein expression and the decrease in CDK4 protein expression (Figure 2E). Cyclin D1, cyclin B1, and CDK2 were not significantly affected in the SAAL1-depleted SK-Hep1 cells when compared to the control cells. Next, we compared the cell cycle progression of SAAL1-depleted cells with the control cells. Flow cytometry analysis revealed that the SAAL1-depleted SK-Hep1 cells showed a significant increase in cell population in the G1 phase and a decrease in the cell population in the S and G2/M phase of the cell cycle as compared to the control cell (Figure 2F). In addition, SAAL1-depleted cells did not show increased annexin V expression on the cell membrane when compared to the control cells (Appendix A), suggesting that SAAL1 knockdown-mediated reduced cell proliferation could not attribute to the increased cellular apoptosis. Together, our findings indicated that inhibition of SAAL1 impaired HCC cell proliferation and 3D colony formation via promoting p21/p27 protein expression and suppressing CDK4 protein to induce G1 phase cell cycle arrest.

### 2.3. Inhibition of SAAL1 Impairs HGF-Induced HCC Cell Migration and Invasion

Hepatocyte growth factor (HGF)/Met axis has been demonstrated to promote tumor growth and metastasis of HCC and high expression of Met has been shown to correlate with a significantly shorter five-year survival in HCC patients [19,20]. Based on the above observations, we then examined if SAAL1 depletion could affect migration and invasion abilities in HGF-stimulated HCC cells [19,21]. The results of the Transwell migration and invasion assay showed that the migration and invasion abilities of SK-Hep1, Hep3B, and PLC/PRF5 cells transfected with SAAL1 siRNA decreased compared to the control groups (Figure 3A,B). When under the stimulation of 50 ng/mL HGF, increased migration and invasion abilities of three HCC lines were observed. The depletion of SAAL1 abolished the HGF-induced HCC cell migration and invasion (Figure 3A,B). Our data revealed that inhibition of SAAL1 expression impaired migration and invasion abilities in HCC cells especially under HGF-stimulation.

### 2.4. Depletion of SAAL1 Significantly Inhibits the HGF/Met/Akt/mTOR Oncogenic Signaling Cascade in HCC

Next, we investigated the possible role of SAAL1 in HGF/Met-driven oncogenic pathways in HCC [20]. One million SK-Hep1 cells were transfected with 10 nM SAAL1 siRNA or the equivalent concentration of control siRNA respectively for 48 h followed by 50 ng/mL HGF treatment for 20 min. Cells were then harvested and subjected to Western blotting analysis. The result of Western blotting analysis showed that HGF-induced Met phosphorylation was slightly impaired in the SAAL1-depleted cells compared to the control cells (Figure 4A). The depletion of SAAL1 significantly inhibited the phosphorylation of Akt and mTOR, which are the downstream effectors of Met-driven oncogenic signaling (Figure 4A). We then investigated whether the depletion of SAAL1 could abolish HGF-induced downstream target gene expression. Osteopontin (OPN) has been characterized as one of the most important HGF-induced downstream targets that could promote HCC invasion and proliferation [22,23,24]. Therefore, we examined whether the inhibition of SAAL1 expression could abolish HGF-induced OPN expression. First, SK-Hep1 cells were transfected with the control siRNA and SAAL1 siRNA, respectively, followed by a 6-h 50 ng/mL HGF treatment. The mRNA expression of OPN was examined by qRT-PCR. The result showed that the depletion of SAAL1 expression could significantly abolish HGF-induced OPN mRNA expression, suggesting that SAAL1 plays an important role in mediating HGF-driven gene expressions (Appendix A). In addition, the subcellular localization of SAAL1 was also investigated. The result of protein fractionation showed that SAAL1 was predominantly presented in the cytosolic fraction and a small portion of membranous SAAL1 was also found in the membrane fraction in SK-Hep1 cells (Figure 4B).

Given that SAAL1 depletion significantly impaired the HGF-driven downstream phosphorylation cascade, we next examined the possibility of whether SAAL1 could physically interact with the key kinases such as Met and mTOR in the HGF-driven oncogenic pathway. To do so, the co-immunoprecipitation (co-IP) experiment was carried out. Western blot analysis of the co-IP experiment showed that Met and SAAL1 were successfully pulled down by an anti-Met and anti-SAAL1 antibody, respectively (Figure 4C). We found that Met proteins were not co-immunoprecipitated with SAAL1 proteins regardless of HGF treatment, and vice versa (Figure 4C). Meanwhile, we observed that mTOR proteins were co-immunoprecipitated with SAAL1 proteins under the condition where SK-Hep1 cells were stimulated with 50 ng/mL HGF for one hour while this physical interaction could not be seen under the condition where cells were not treated with HGF (Figure 4C). However, the physical association between mTOR and SAAL1 could be weak and transient due to few mTOR proteins being co-immunoprecipitated with SAAL1 proteins.

Taken together, our data showed that SAAL1 could play a pivotal role in the HGF-regulated oncogenic pathway through interacting with mTOR in HCC.

### 2.5. Inhibition of SAAL1 Significantly Increases Chemosensitivity towards Sorafenib and Foretinib Treatment in HCC Cells

Chemoresistance is the main hindrance in the treatment of advanced HCC with a relatively low five-year survival rate [3,5]. Multi-kinase inhibitors such as sorafenib have emerged as a new class of drug for the treatment of advanced HCC patients to improve their survival rate [5,9,25,26]; however, drug resistance or intolerance to sorafenib is frequently seen [26,27]. The main mechanisms of drug resistance to sorafenib include the activation of PI3K/Akt or Met signaling pathways [27,28,29,30,31]. Therefore, the Met kinase inhibitor, such as foretinib, has been suggested as a treatment option after sorafenib resistance in HCC patients [25,32]. Our data showed that inhibition of SAAL1 expression could lead to a reduction in HGF-induced PI3K/Akt/mTOR phosphorylation cascade. Therefore, we investigated if the depletion of SAAL1 could increase the chemosensitivity of HCC cells towards sorafenib and foretinib.

First, the effect of SAAL1 depletion on HCC cell proliferation in the presence or absence of sorafenib/foretinib treatment was examined. The MTS assay showed that SAAL1-depleted SK-Hep1 cells without treatment with sorafenib/foretinib showed decreased cell proliferation rate compared to the control cells (Figure 4D,E). As expected, treatment of SK-Hep1 cells with 5 μM sorafenib or 2.5 μM foretinib significantly decreased the cell proliferation rate compared to the cells without sorafenib/foretinib treatment. Moreover, we found that SAAL1-depleted cells treated with 5 μM sorafenib or 2.5 μM foretinib showed a further impaired cell proliferation rate when compared with the control cells treated with the same condition (Figure 4D,E).

We then conducted an in vitro cytotoxicity assay and found that SAAL1-depleted SK-Hep1 cells showed a two-fold decrease in the IC_50_ value under sorafenib treatment (IC_50_ = 3.796 μM) compared to the control cells (IC_50_ = 8.448 μM) (Figure 4F). Similarly, SAAL1-depleted SK-Hep1 cells showed approximately a nine-fold decrease in IC_50_ under foretinib treatment (IC_50_ = 0.301 μM) compared to the control cells (IC_50_ = 2.756 μM) (Figure 4G). These data suggested that inhibition of SAAL1 increased the drug sensitivity of SK-Hep1 cells towards sorafenib or foretinib treatment. Taken together, our data revealed that inhibition of SAAL1 could increase the chemosensitivity of HCC cells towards sorafenib or foretinib treatment, and SAAL1 siRNA may exert synergistic effects with sorafenib or foretinib in the treatment of HCC.

## 3. Discussion

HCC remains a devastating disease with rapidly growing morbidity and mortality in recent years [13]. Although the treatment of advanced HCC has evolved with several new multi-kinase inhibitors and immunotherapy to improve survival, the treatment effect of advanced HCC is still unsatisfactory [12]. In this study, we unveiled the novel oncogenic role of SAAL1 and its clinical impact on overall survival in HCC patients for the first time. We are the first to indicate that the depletion of SAAL1 affected the characteristics of cancer aggressiveness and inhibited the HGF/Met-driven Akt/mTOR signaling pathway. We found that suppression of SAAL1 significantly increased chemosensitivity towards sorafenib or foretinib treatment in HCC cells. The synergistic effect of SAAL1 siRNA with sorafenib or foretinib was also demonstrated in the treatment of HCC cells. SAAL1 could potentially serve as a prognostic biomarker and a promising target for developing HCC therapeutics.

As an SAA related gene, SAAL1 was also identified as an acute phase reactant upon liver injury and inflammation [14]. Although no information regarding SAAL1 and cancer was reported before, SAA was reported to be produced within tumor tissues including colorectal, ovarian, and uterine cancer [17]. SAA family genes were also found to be involved in promoting angiogenesis and metastasis [17]. Our data showed that SAAL1 was significantly upregulated in HCC tumor tissues compared to the normal tissues. The depletion of SAAL1 was found to impair HGF-stimulated cell migration and invasion significantly. These findings are in accordance with previous reports in SAA [17]. Moreover, inhibition of SAAL1 also inhibited the colony-forming abilities of HCC lines in a 3D soft agar culture, suggesting that SAAL1 plays a critical role in the anchorage-independent growth of HCC cells.

Hepatocarcinogenesis consists of complex genetic alterations involving multiple signaling pathways including epidermal growth factor (EGF), vascular endothelial growth factor (VEGF), Ras mitogen-activated protein kinase (MAPK), HGF/C-Met, PI3K/PTEN/Akt/mTOR, and Wnt/β-Catenin pathways [6]. Our data entailed that SAAL1 may be associated with HGF/Met signaling because inhibition of SAAL1 decreased HGF-stimulated cell migration and invasion. Our further investigation showed that HGF-induced activation of HGF/Met downstream effectors such as Akt and mTOR were significantly impaired in the SAAL1-depleted cells compared to the control cells (Figure 4A). Moreover, the co-IP experiments showed that SAAL1 could interact with mTOR in HCC cells under the condition where the Met/Akt/mTOR signaling axis is activated by HGF (Figure 4C). Together, our study demonstrated that SAAL1 plays an important role in mediating HGF/Met-driven Akt/mTOR oncogenic signaling and downstream target gene expression via interacting with mTOR in HCC.

Sorafenib was the first multi-kinase inhibitor approved by the U.S. Food and Drug Administration (FDA) for the treatment of advanced HCC in 2007 [13]. Despite an initial significant response to prolong overall survival in HCC patients, resistance to sorafenib via several escape or compensatory mechanisms including activation PI3K/Akt or Met signaling were reported before [9,26,27,28,29,30,31]. Amplification of HGF/Met axis to promote HCC cell proliferation and metastasis also renders Met as a therapeutic target in HCC [19,20], hence Met inhibitors such as foretinib and tivantinib were also applied in the treatment of HCC [5,19,26,29]. In our study, we found that inhibition of SAAL1 expression significantly impaired the HGF/Met-driven PI3K/Akt/mTOR signaling pathway, and depletion of SAAL1 can restore chemosensitivity of HCC cells towards both sorafenib and foretinib treatment. These findings suggested that targeting SAAL1 may serve as a promising strategy to overcome sorafenib resistance. We also found that SAAL1 siRNA showed a synergistic effect in combination with sorafenib and foretinib in the treatment of HCC cells. Future study will focus on designing a liposome-based system for delivery of SAAL1 siRNA in combination with sorafenib or foretinib to observe antitumor efficacy in an orthotopic HCC mouse model [33].

## 4. Materials and Methods

### 4.1. Cell Culture

Three human HCC lines, SK-Hep1, Hep3B, and PLC/PRF5 were obtained from the American Type Culture Collection (ATCC). All HCC lines were cultured in Dulbecco’s Modified Eagle Medium (DMEM) supplemented with 10% fetal bovine serum (FBS) (Invitrogen, Carlsbad, CA, USA). Cells were incubated in a humidified incubator at 37 °C with 5% CO_2_. All cell lines were validated as mycoplasma-free on 20 January 2019, using DAPI staining and are routinely examined by DNA STR for authentication.

### 4.2. Transwell Migration and Invasion Assay

The detailed procedures were described previously [34]. In brief, 1 × 10^5^ cells were seeded in the 6 μm Transwell insert (BD Biosciences, San Jose, CA, USA) sitting on the 24-well plate. Next, 0.5 mL DMEM with 10% FBS was added to the 24-well plate. After 24-h incubation, the migrated cells were stained with 0.5% crystal violet staining solution for 20 min. For the invasion assay, all the procedures were the same except the Matrigel-coated Transwell insert (BD Biosciences) was used to replace the 6-μm Transwell insert. The migrated/invaded cells were quantified using ImageJ software (NIH, Bethesda, MD, USA).

### 4.3. Western Blot

Cells were harvested in RIPA buffer (1% TritonX-100, 50 mM pH:7.4 Tris-HCl, 150 mM NaCl_2_, and 0.1% SDS) and were subjected to SDS-PAGE electrophoresis. Protein was transferred onto methanol-activated PVDF membrane at 55 V for three hours. PVDF membrane was washed twice with 1× TBST and blocked in 5% non-fat milk for one hour at room temperature. The membrane was then hybridized with the specific primary antibodies followed by the HRP-conjugated secondary antibody.

### 4.4. Antibodies and Chemical Reagents

Anti-SAAL1 antibody and anti-Lamin B2 antibody were purchased from Bethyl and Abcam. Anti-phospho-Met antibody, anti-Met antibody, anti-phosphor-mTOR antibody, anti-mTOR antibody, anti-phosphor-Akt antibody, anti-Akt antibody, and anti-Na and K-ATPase antibody were purchased from Cell Signaling Technology. Anti-α-Tubulin antibody, anti-Actin antibody, and the HRP-conjugated secondary antibodies were purchased from Santa Cruz Biotechnology Inc. Sorafenib and foretinib were purchased from Merk and Selleckchem, respectively. The detailed information on antibodies and reagents are listed in the Appendix A.

### 4.5. Oligonucleotide Transfection

One million HCC cells were seeded in the 6-well dishes 24 h prior to the transfection. SAAL1 siRNAs and the control siRNAs were introduced into HCC cell lines respectively using TransIT-X_2_^®^ system (Mirus, Madison, USA), and the transfected cells were incubated for 48 h. The siRNA sequences used in this study were as follows: SAAL1-1 siRNA: sense: 5′-CCACCUACUCUGCUGGAAATT; anti-sense: 5′-UUUCCAGCAGAGUAGGUGGTT. SAAL1-2 siRNA: sense: 5′-GGUUGUGGACAAGCUCUUUTT; anti-sense: AAAGAGCUUGUCCACAACCTT. The control siRNA: sense: 5′-UUCUCCGAACGUGUCACGUTT; anti-sense: 5′-ACGUGACACGUUCGGAGAATT. The detailed information of siRNA oligonucleotides and transfection reagent are listed in Appendix A.

### 4.6. Soft Agar Assay

A total of 1 mL of the preheated DMEM with 10% FBS containing 0.5% agarose was loaded in the 12-well plate and the plate was placed in the incubator at 37 °C to allow the medium to harden to make the base agar. For the upper layer, 2 × 10^4^ cells were quickly mixed with 1 mL of the preheated DMEM with 10% FBS containing 0.25% agarose (not to exceed 40 °C) and loaded above the bottom layer. The plate was immediately placed back in the 37 °C incubator and was incubated for two weeks. Colonies were stained with 0.05% (wt/vol) iodonitrotetrazolium chloride (Sigma, St. Louis, MO, USA) for two days before the endpoint. The number of colonies was quantified using ImageJ software (NIH, USA).

### 4.7. MTS Cell Proliferation Assay

Seeded in the 96-well plate was 3 × 10^3^ cells for 24 h prior to the MTS assay. Cell viability was analyzed on the indicated time point using the CellTiter96^®^ MTS assay kit (Promega, Madison, WI, USA) according to the instructions from the manufacturer. All the experiments were performed in triplicate and repeated three times.

### 4.8. In Vitro Cytotoxicity Assay

Cells (with a density around 6 × 10^3^ per well) were seeded in the 96-well plate 24 h prior to the drug treatment. Cells were then treated with sorafenib or foretinib with the indicated dosage for 48 h. Cell viability was determined using the CellTiter96^®^ cell proliferation assay kit (Promega, CA, USA) according to the instructions from the manufacturer. The IC_50_ value of cells treated with or without sorafenib or foretinib was calculated using GraphPad Prism software (San Diego, CA, USA). All the experiments were performed in triplicate and repeated three times.

### 4.9. Protein Subcellular Fractionation

Protein subcellular fractionation was performed using the Cell Fractionation kit according to the manufacturer’s instructions (Cell Signaling Technology, Danvers, MA, USA). Briefly, cells were washed with cold 1× PBS and scratched to harvest the pellets. The pellets were lysed in 500 uL cytoplasmic isolation buffer (CIB) and incubated on ice for 5 min. The lysate was centrifuged for 5 min at 500× *g* at 4 °C. The supernatant was the cytoplasmic fraction. The pellets were resuspended with 500 uL membrane isolation buffer (MIB) and incubated on ice for 5 min. The lysate was centrifuged for 5 min at 500× *g* at 4 °C. The supernatant was the membrane fraction. The pellets were resuspended with cytoskeleton/nucleus isolation buffer and sonicated for 5 s three times. The resulting fraction was the cytoskeleton/nuclear fraction. Three subcellular fractions were resolved in 10% SDS-PAGE followed by Western blotting analysis.

### 4.10. Co-Immunoprecipitation

In a 15-cm culture dish, 1 × 10^7^ SK-Hep1 cells were grown followed by serum starvation for 16 h before HGF stimulation. The serum-starved cells were stimulated with 50 ng/mL HGF for one hour and harvested in co-IP buffer (0.5% NP-40, 25 mM Tris-HCl, 150 mM NaCl, and 2 mM EDTA). One milligram of protein lysate was pre-incubated with a 30 μL protein G Sepharose^®^ Xtra magnetic bead (Cytiva, WA. USA) at 4 °C for one hour to remove the potential non-specific binding proteins existing in the protein samples. The protein-bead mixtures were centrifugated at 1500 rpm for 5 min and the supernatants containing protein lysate were transferred to the new tubes for the subsequent co-IP experiment. Anti-Met antibody and anti-SAAL1 antibody at 2 μg were added to the protein lysate, respectively and the protein-antibody mixtures were incubated at 4 °C in a rotating apparatus overnight. Next, 30 μL protein G magnetic beads were added to the protein-antibody mixtures and incubated at room temperature for an additional 2 h. The samples were centrifuged to remove the supernatant containing unbound proteins and the magnetic beads were washed three times with co-IP buffer. The beads were then resuspended in 1X sample buffer and boiled at 95 °C for 3 min followed by Western blot analysis.

### 4.11. Cell Cycle Analysis

The SK-Hep1 cells were transfected with SAAL1 siRNA- and the control siRNA respectively for 48 h followed by 16-h serum starvation. Cells were detached by trypsin followed by fixation with 70% ethanol. The fixed cells were treated with 200 μg/mL propidium iodide (PI)(Sigma, CA, USA) for 30 min. The cell cycle analysis was carried out using BD FACSCanto™ flow cytometry. Data were processed using FlowJo™ v10 software.

### 4.12. Annexin V Staining

The SK-Hep1 cells were transfected with SAAL1 siRNA and the control siRNA, respectively for 48 h. Cells were detached by trypsin followed by fixation with 70% ethanol. The fixed cells were stained with FITC annexin V (BD Biosciences, CA, USA) for 30 min. FITC annexin V positive cells were analyzed using BD FACSCanto™ flow cytometry. Data were processed using FlowJo™ v10 software (v10, BD Biosciences, CA, USA).

### 4.13. Public Domain Dataset Analysis

The level 3 FPKM (fragments per kilobase per million mapped reads)-normalized RNA-seq data containing 347 liver cancer cases and 50 normal cases along with their clinicopathological information and survival data were retrieved from The Cancer Genome Atlas (TCGA) (https://cancergenome.nih.gov) for statistical analysis. Furthermore, 346 liver cancer cases from TCGA with the survival information were used for Kaplan-Meir survival analysis. In addition, SAAL1 mRNA expressions of 194 liver cancer cases and 50 normal cases were retrieved from the on-line database “Gene Expression across Normal and Tumor tissue (GENT)” (http://medicalgenome.kribb.re.kr/GENT) for statistical analysis.

### 4.14. Statistical Analysis

The Student’s t-test was used for the comparison between the control and the experimental groups. The cut-off value of SAAL1 using receiver-operating curve (ROC) analysis was applied to separate 346 HCC patients into the high SAAL1 group and the low SAAL1 group. Overall survival results were presented as a Kaplan-Meier survival curve based on the cut-off value, and the log-rank test was applied to estimate the statistical significance. Multivariate Cox’s regression analysis was used to determine the independent predictor of overall survival using a variable (SAAL1 expression) significant in univariate analysis as a covariate. All data were presented as mean ± SEM (standard error of the mean). Differences with *p* < 0.05 (* *p* < 0.05, ** *p* < 0.01 and *** *p* < 0.001) were considered statistically significant.

## 5. Conclusions

Our study sheds light on the novel role of SAAL1 as an oncogenesis regulator in HCC and helps to better understand the possible mechanisms of sorafenib resistance regulated by SAAL1 in HCC. SAAL1 may serve as a useful prognostic biomarker, as well as a potential target in the treatment of HCC.

## Figures and Tables

**Figure 1 cancers-12-01843-f001:**
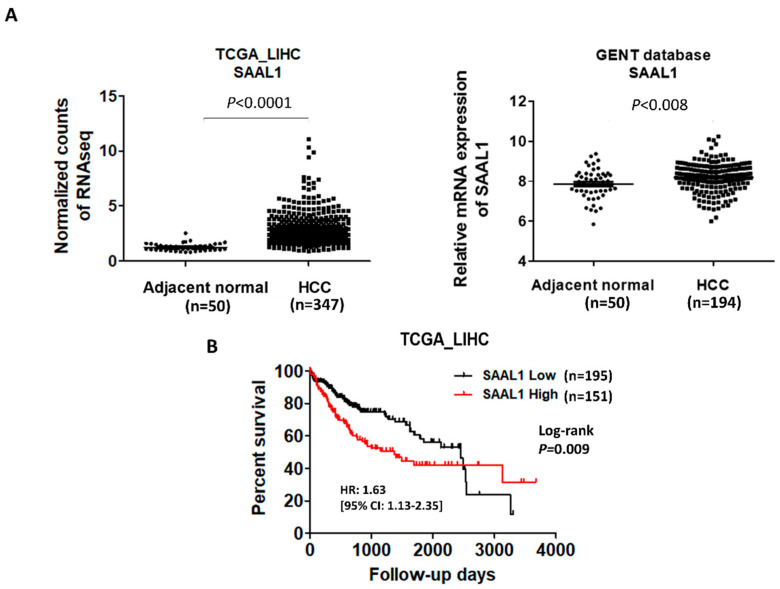
The expression level of SAAL1 increases in HCC tumor tissues and correlates with poor overall survival in HCC patients. (**A**) Analysis of the expression level of SAAL1 in HCC patients using TCGA and GENT databases. (**B**) Kaplan-Meier survival analysis of HCC patients according to SAAL1 RNAseq data retrieved from TCGA dataset.

**Figure 2 cancers-12-01843-f002:**
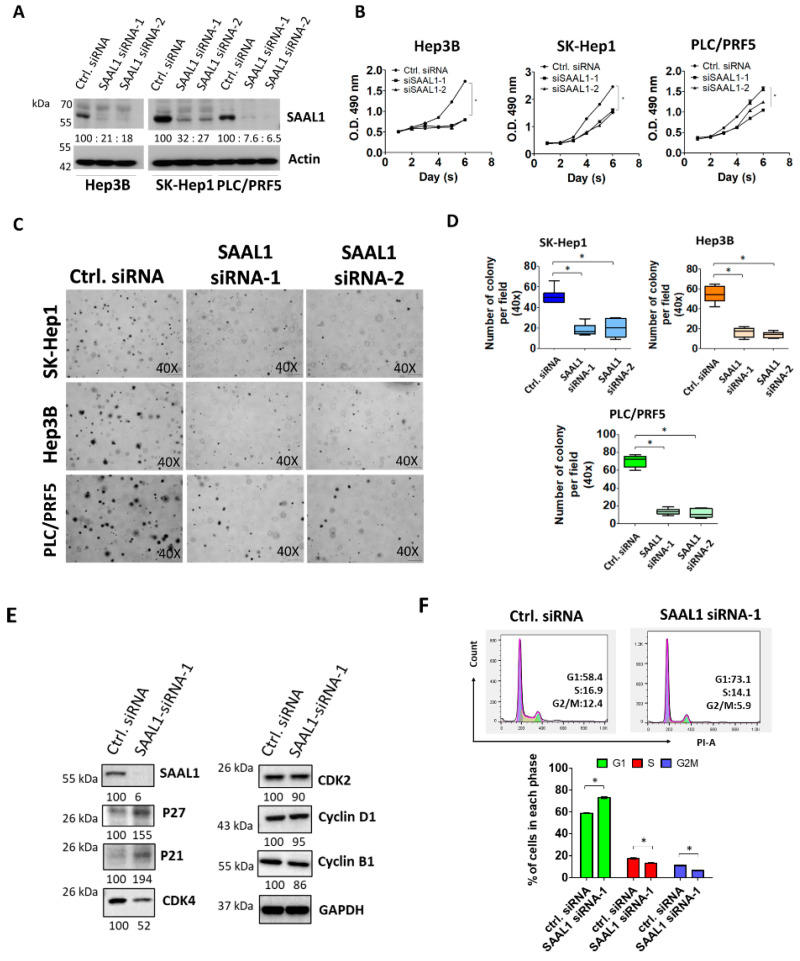
Depletion of SAAL1 expression impairs cell proliferation and 3D colony formation via inducing G1-phase cell cycle arrest. (**A**) Western blotting analysis of SAAL1 protein expression in three HCC lines transfected with SAAL1 siRNAs. Actin was served as an internal control. (**B**) Depletion of SAAL1 reduces cell proliferation of HCC cells. * *p* < 0.05. (**C**) Inhibition of SAAL1 expression reduces the colony-forming abilities of HCC cells in a 3D soft agar culture. (40×, brightfield). (**D**) The quantitative results of the 3D soft agar assay. (**E**) Western blotting analysis of cell cycle proteins in SAAL1-depleted SK-Hep1 cells and the control SK-Hep1 cells. (**F**) Flow cytometry analysis of cell cycle progress in SAAL1-depleted cells and the control cells. * *p* < 0.05. Each experiment was performed in triplicate and was repeated three times. The representative data were shown as pictures.

**Figure 3 cancers-12-01843-f003:**
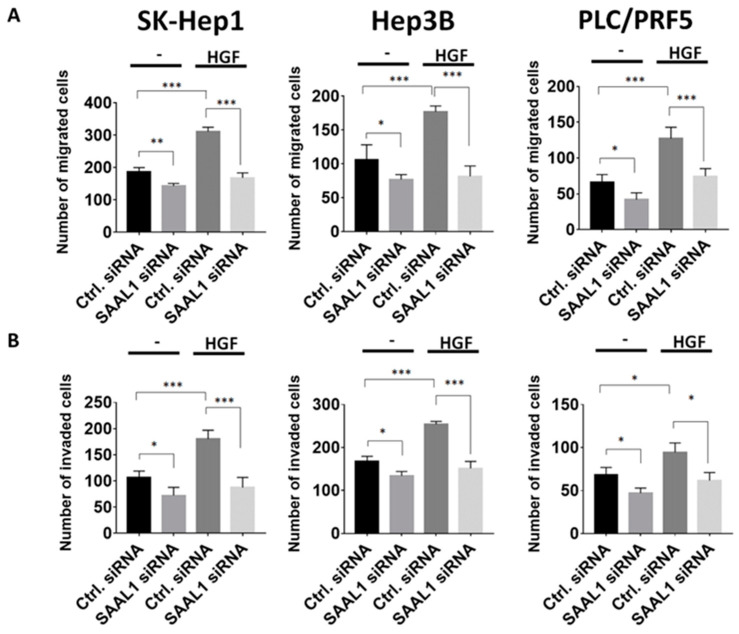
Inhibition of SAAL1 expression suppresses HGF-induced HCC cell migration and invasion. (**A**) The effect of SAAL1 depletion on HGF-induced HCC cell migration was determined using the Transwell system. (**B**) The effect of SAAL1 depletion on HGF-induced HCC cell invasion was determined using the matrigel-coated Transwell system. Quantitative data are shown by histograms with means ± SEM (*n* = 3). * *p* < 0.05, ** *p* < 0.01, *** *p* < 0.001. All experiments were performed in triplicates and were done at least three times.

**Figure 4 cancers-12-01843-f004:**
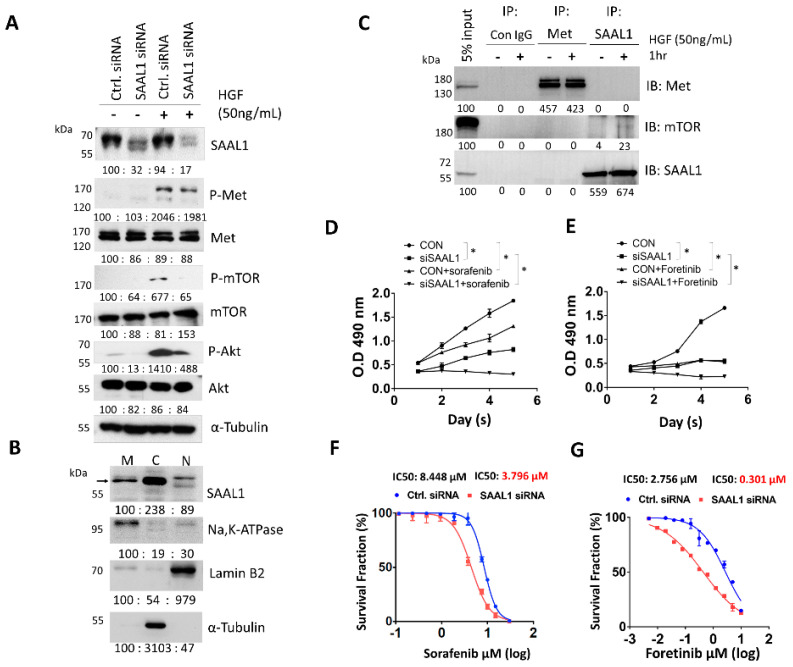
SAAL1 depletion sensitizes HCC cells to sorafenib and foretinib treatment through impairing HGF/Met-driven Akt/mTOR oncogenic pathway. (**A**) The effect of SAAL1 depletion on the HGF-induced Met/Akt/mTOR phosphorylation cascade was analyzed by Western blotting analysis, α-tubulin was served as an internal control. (**B**) Subcellular localization of SAAL1 protein in SK-Hep1 cells was analyzed by Western blotting analysis, α-tubulin was served as an internal control. (**C**) Analysis of SAAL1-interacting proteins in the absence or presence of HGF stimulation. The control IgG (Con IgG) was used as a negative antibody control. A total of 25 μg protein lysates were loaded and served as an input control for the co-IP experiment. SAAL1 and Met were immunoprecipitated with the specific IP antibody, respectively. (**D**) The effect of sorafenib treatment on cell proliferation of SAAL1-depleted SK-Hep1 cells versus the control SK-Hep1 cells. Data are means ± SEM (*n* = 3). * *p* < 0.05. (**E**) The effect of foretinib treatment on cell proliferation of SAAL1-depleted SK-Hep1 cells versus the control SK-Hep1 cells. Data are means ± SEM (*n* = 3). * *p* < 0.05. (**F**) Determination of the IC_50_ value of sorafenib in SAAL1-depleted SK-Hep1 cells versus the control cells. (**G**) Determination of the IC50 value of foretinib in SAAL1-depleted SK-Hep1 cells versus the control cells. All experiments were performed in triplicates and were done at least three times.

**Table 1 cancers-12-01843-t001:** Univariate and multivariate Cox’s regression analysis of SAAL1 gene expression for overall survival of 346 patients with HCC.

Characteristic	No. (%)	OS			
		CHR (95% CI)	*p*-Value	AHR (95% CI)	*p*-Value
SAAL1	(*n* = 346)				
Low	195 (56.4)	1.00		1.00	
High	151 (43.6)	1.63 (1.13–2.35)	0.009	1.57 (1.09–2.27)	0.016

Abbreviation: OS, overall survival; CHR, crude hazard ratio; AHR, adjusted hazard ratio; AHR were adjusted for AJCC pathological stage (II, III and IV VS. I).

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
