# Peer review of "Identification of the Novel Oncogenic Role of SAAL1 and Its Therapeutic Potential in Hepatocellular Carcinoma"

_cancers, 2020, doi:10.3390/cancers12071843_

Round 1

Reviewer 1 Report

The authors have adequately addressed the questions raised by the reviewer of the previous submission by providing new data and modifying the manuscript. 

Reviewer 2 Report

This manuscript is well written and have promising results.

This manuscript is a resubmission of an earlier submission. The following is a list of the peer review reports and author responses from that submission.

Round 1

Reviewer 1 Report

In this manuscript the authors report the hepatocarcinogenic properties of SAAL1, an acute phase protein. They showed that SAAL1 was significantly up-regulated in HCC tumor tissues compared to the adjacent benign tissues and high expression of SAAL1 correlated with poor overall survival. Their data also showed that SAAL1 knock down reduced HCC cell proliferation in monolayers and in 3D cultures by blocking  with impaired HGF/Met- and its downstream Akt/mTOR signally.  Furthermore, SAAL1 depletion enhanced  sensitivity of HCC cells to sorafenib and foretinib.

Overall, well written manuscript and the data are well presented. However, there are some key issues.

  1. The authors have not convincingly demonstrated that the tumor-promoting function of SAAL1 is through activation of HGF/MET signaling pathway. Membrane localization of SAAL1 and its direct interaction with MET data are not convincing. Membrane localization of SAAL1 data using biochemical fractionation (Fig 4B) is not strong. It has to be confirmed by immunofluorescence and co-immunoprecipitation analyses.  Furthermore, rescue experiments are needed to confirm the role of MET in mediating SAAL1’s function. Does expression of HGF/ MET downstream target  genes modulated by SAAL1 overexpression and depletion?
  2. In addition to SAAL1 protein level, depletion of SAAL1 RNA level in siRNA-1/2 depleted cells has to demonstrated.  

Minor comments

  1. Describe the nature of Oplegnathus fasciatus model (line 52, page 2).
  2. The rationale for selecting foretinib is not provided.
  3. In line 100, page 2 “control cells” should be control siRNA.
  4. A reference for GENT database is needed.
  5. PLC-5 should be PLC/PRF5
  1. Spelling correction is needed.
  2. The figure subpanel labels in the text are in lower case but in uppercase in the figures.

Author Response

We appreciate the time and efforts of the reviewer in reviewing and commenting on our manuscript. We have performed additional experiments as suggested and revised the manuscript according to the reviewer's comments and questions. Our point to point response letter is attached.

Best regards,

Shih-Hsuan Chan

Reviewer 2 Report

In this manuscript, the authors identified the novel oncogenic role of SAAL1 and high expression of SAAL1 correlated with shorter OS. They also presented that the depletion of SAAL1 significantly reduced cell proliferation and increased chemosensitivity of HCC cells in vitro study. SAAL1 is fascinating molecule and can be one of the targets in cancer therapy. This manuscript is interesting and well-written, but it still has some problems as indicated below.

Major

  1. The authors concluded that HCC patients with high expression of SAAL1 correlated with a significant shorter OS than those with low SAAL1 expression (Figure 1B). In this analysis, they have to show clinical backgrounds of these two groups. The authors also have to present the predict factors of OS by multivariate analysis, and show that one of these factors is SAAL1 expression.  
  2. The authors showed that depletion of SAAL1 expression impairs cell proliferation. This might be true, but its mechanism is unclear. Did it cause cell cycle arrest or to increase cell apoptosis. The authors should do experiments to clarify the mechanism.  
  3. The authors concluded that SAAL1 plays an important role in regulating HGF/MET-driven Akt/mTOR signaling. How does this molecule of SAAL1 activate this signaling cascade? P-mTOR was inhibited by knockdown of SAAL1 siRNA, and this result shows SAAL1 might be involved the upstream of this signaling pathway. The authors explained that this result provided first evidence, so it should be investigated in detail, for example, SAAL1 might works as a cofactor or bind to intracellular domain of mTOR.
  4. The authors showed that inhibition of SAAL1 increases chemosensitivity towards sorafenib treatment in vitro assay. If expression of SAAL1 is enhanced by sorafenib, this result might be valuable in clinical setting. If not, this is not main reason of chemoresisitant to sorafenib.

Author Response

(The authors gave the same response as above.)
